Insights into the metagenomic and metabolomic compositions of the bacterial communities in Thai traditional fermented foods as well as the relationships between food nutrition and food microbiomes

Nimnoi Pongrawee 1
http://orcid.org/0000-0001-8691-7036 Pongsilp Neelawan 2 pongsilp_n@su.ac.th
1 Department of Science and Bioinnovation, Faculty of Liberal Arts and Science, Kasetsart University , Nakhon Pathom , Thailand
2 Department of Microbiology, Faculty of Science, Silpakorn University , Nakhon Pathom , Thailand
Grohmann Elisabeth
Electronic publication date: 2025 Jun 27
Publication date: 2025
Volume: 13
Electronic Location ID: e19606
Received 2025 Jan 17; Accepted 2025 May 23
Copyright: © 2025 Nimnoi and Pongsilp
Copyright year: 2025
Copyright holder: Nimnoi and Pongsilp
License: This is an open access article distributed under the terms of the Creative Commons Attribution License, which permits unrestricted use, distribution, reproduction and adaptation in any medium and for any purpose provided that it is properly attributed. For attribution, the original author(s), title, publication source (PeerJ) and either DOI or URL of the article must be cited.
License URL: https://creativecommons.org/licenses/by/4.0/

Keywords: Bacterial community structure, Fermented food, Food nutrition, Illumina next-generation sequencing, Metabolic map, Thai traditional food, Biomarker

Funding: Faculty of Science, Silpakorn University SRIF-JRG-2567-05 This work was financially supported by Faculty of Science, Silpakorn University, under grant number SRIF-JRG-2567-05. The funders had no role in study design, data collection and analysis, decision to publish, or preparation of the manuscript.

==============================
Five Thai traditional fermented foods, including khao-mak (sweet fermented sticky rice), pak-kard-dong (sour salt-fermented mustard greens), nor-mai-dong (sour salt-fermented bamboo sprouts), moo-som (sour salt-fermented pork), and pla-som (sour salt-fermented fish), were analyzed for their food nutrition and bacterial community structures. Sour salt-fermented bamboo sprouts possessed the highest unique amplicon sequence variables (ASVs), which were 3,476, as well as the highest bacterial diversity and richness, while in contrast, sweet fermented sticky rice possessed the lowest such indices. The phylum Firmicutes accounted for the largest proportions in both sour salt-fermented meats and sweet fermented sticky rice whereas the Proteobacteria occupied the largest proportions in both sour salt-fermented vegetables. The bacterial community structures of both sour salt-fermented meats were similar in terms of composition at class level, while the dominant genera compositions were totally different among all foods. Gene functions, enzymes, and metabolic pathways annotated from the bacterial communities in all foods were those involved in growth metabolisms, genetic information processing, environmental information processing, and cellular signaling. Sour salt-fermented bamboo sprouts had the highest numbers of unique annotated genes, enzymes, and metabolic pathways.

Introduction

Fermented foods are defined as foods made through desired microbial growth and enzymatic conversions of raw food ingredients (Marco et al., 2021). Traditional fermented foods are prepared and consumed by the regional populations around the world since ancient times, and these foods are recognized for many health benefits. Numerous beneficial microorganisms impart unique properties to fermented foods through their metabolic activities, therefore the analyses of microbial diversity and abundance in fermented foods will facilitate understanding of the roles of microbial taxa in establishing organoleptic properties and offering health benefits (Deka et al., 2021). During the fermentation process, microbial community converts raw food ingredients into products which improve particular organoleptic properties, shelf life, and nutraceutical quality of final fermented foods (Xing et al., 2023; Zhang et al., 2023).

Thailand is a humid tropical country with diverse ecosystems. Thus, there is a wide range of agricultural commodities which are raw materials for many Thai traditional fermented foods. Here are examples of the popular ones which were employed in this study. Khao-mak is sweet fermented sticky rice with low alcohol content. For its preparation, steam-cooked sticky rice is mixed with flour balls containing natural starter microorganisms, and kept in a closed container for 2–3 days. Enzymes produced by starter microorganisms hydrolyze starch in sticky rice into sugar, some parts of which are subsequently fermented to alcohol and organic acids (Rittisorn et al., 2024). Pak-kard-dong is sour salt-fermented mustard greens prepared by soaking mustard greens in brine for a day and then submerging them under fermented water containing soluble salt, soluble sugar, and rice washing water, in a closed container for 4–5 days. Nor-mai-dong is sour salt-fermented bamboo sprouts prepared by soaking peeled bamboo sprouts in brine for a day, and then submerging them under water in a closed container for 7 days. Moo-som is sour salt-fermented pork whose ingredients are pork, salt, cooked rice, garlic, and pepper. All ingredients are crushed together to form a paste which is then tightly wrapped in a plastic bag for 2–3 days. Pla-som is sour salt-fermented fish whose ingredients are fish, salt, cooked rice, garlic, and pepper. Fish is soaked in rice washing water containing soluble salt and then soaked in brine. All ingredients are crushed together to form a paste which is then tightly wrapped in a plastic bag for 2–3 days. The spontaneous fermentation of fermented foods engenders the proliferation of numerous microorganisms. Among them, the lactic acid bacteria (LAB) mainly contribute to the accumulation of lactic acid, which inhibits the growth of harmful microorganisms by lowering pH, and also provides palatable flavors (Du et al., 2022; Lv et al., 2021). Up to date, many genera have been reported as the LAB Firmicutes (Abedi & Hashemi, 2020; Hu et al., 2023; Wang et al., 2021a). They collectively belong to the phylum Firmicutes, class Bacilli, orders Bacillales and Lactobacillales, and families Bacillaceae, Enterococcaceae, Lactobacillaceae, and Streptococcaceae according to the National Center for Biotechnology Information (NCBI) Taxonomy database (https://www.ncbi.nlm.nih.gov/taxonomy/). Different microbial communities engender different complex biochemical and physical reactions, establishing diverse and distinct organoleptic properties of final fermented foods (Alkema et al., 2016). Thus, the bacterial diversity and community merit the evaluation of their roles in establishing and controlling organoleptic properties of fermented foods.

The recent advances in next-generation sequencing (NGS) technologies have provided the information on microbial diversity and communities in fermented foods, although there is still a lack of clarity on food nutrition, specific bacterial taxa, gene functionality, enzymes, and metabolic pathways which are compared across various types of food fermentation. Therefore, the objectives of this study were to characterize the nutritional compositions as well as the bacterial diversity and communities of five Thai traditional fermented foods including khao-mak (sweet fermented sticky rice), pak-kard-dong (sour salt-fermented mustard greens), nor-mai-dong (sour salt-fermented bamboo sprouts), moo-som (sour salt-fermented pork), and pla-som (sour salt-fermented fish). The food nutrition parameters, including calories, carbohydrate, fat, moisture, protein, total sugar, and sodium contents, were determined. The bacterial diversity and communities were analyzed using an Illumina NGS platform. Metagenomic and metabolomic analyses were employed to annotate the gene functions, enzymes, and metabolic pathways. The correlations between food nutrition parameters and bacterial taxa were also evaluated. To the best of our knowledge, this is the first report investigating the following aspects of Thai traditional fermented foods: (1) the food nutrition facts; (2) the bacterial diversity and communities; (3) the annotated gene functions, enzymes, and metabolic pathways; and (4) the influences of food nutrition parameters on abundance of bacterial taxa. This ultimate data would be beneficial for quality control of the bacterial succession in the fermentation processes.

Materials and Methods

Thai traditional fermented foods

Five Thai traditional fermented foods, including Khao-Mak (sweet fermented sticky rice; sample A), pak-kard-dong (sour salt-fermented mustard greens; sample B), nor-mai-dong (sour salt-fermented bamboo sprouts; sample C), moo-som (sour salt-fermented pork; sample D), and pla-som (sour salt-fermented fish; sample E), were purchased from Boon Mueang fresh market in Mueang district, Lopburi province of Thailand (14°48′13.662″N 100°36′38.636″E) on 21 January 2024. Food samples were placed in sterile plastic bags, immediately stored in ice boxes, and delivered to the laboratory within 24 h after purchase.

Food nutritional analyses

All food samples (A–E) were determined for their nutrition facts. Calories and carbohydrate contents were analyzed using the in-house method TE-CH-169 based on the method of analysis for nutrition labeling (Sullivan & Carpenter, 1993). Fat content was analyzed using the Association of Official Analytical Chemists (AOAC) method 922.06 (Association of Official Analytical Chemists (AOAC), 2023). Moisture content was analyzed using the AOAC methods 925.45A (for samples A, B, and C) and 950.46B (for samples D and E) (Association of Official Analytical Chemists (AOAC), 2023). Protein, total sugar, and sodium contents were determined by the In-house method TE-CH-042 based on the AOAC method 981.10, the In-house method TE-CH-164 based on the AOAC method 977.20, and the In-house method TE-CH-134 based on the AOAC method 984.27, respectively (Association of Official Analytical Chemists (AOAC), 2023).

Food DNA extraction, illumina NGS, data processing, and bioinformatic analyses

To explore the bacterial communities in five Thai traditional fermented foods through an Illumina NGS, total bacterial DNA was extracted from food samples by using a DNeasy mericon food kit (Qiagen, Inc., Hilden, Germany). DNA samples were extracted from nine fractions of each food sample, then three of them were pooled as one composite DNA sample. Therefore, three composite DNA samples were prepared for each food. The protocols for amplification of the V4 variable region of the 16S rRNA, construction of DNA library, and Illumina sequencing were followed as described in the previous article (Pongsilp & Nimnoi, 2024). Data of raw sequence reads was converted to fastq files. The fastq files were consecutively processed using the FLASH software version 1.2.11 (https://ccb.jhu.edu/software/FLASH/) (Magoč & Salzberg, 2011), the FASTP software version 0.20.1 (https://github.com/OpenGene/fastp) (Chen et al., 2018), the VSEARCH software version 2.21.1 (https://github.com/torognes/vsearch/releases) (Rognes et al., 2016), and the QIIME2 software version 2021.4 (https://forum.qiime2.org/) (Bolyen et al., 2019) to obtain the effective tags. The effective tags, whose sequence abundance was less than five, were excluded to select the final amplicon sequence variables (ASVs). The Classify-sklearn moduler in the QIIME2 software version 2021.4 was employed to compare the ASVs with the sequences available in the SILVA SSU rRNA database (https://www.arb-silva.de/) (Quast et al., 2013) for species annotation of each ASV. The ASV data has been deposited into the Sequence Read Archive of the NCBI under the BioProject ID: PRJNA1180770.

Statistical analyses

The analyses for statistical processing of Illumina sequencing data were as follows: (1) the computations of the parameters relative to alpha diversity, which included community diversity (Shannon-Weaver and Simpson’s indices), community richness (Chao1 index and number of observed species), sequencing depth index (Good’s coverage), and dominance as well as beta diversity for quantifying sample variation in species complexity; (2) the principal coordinate analysis (PCoA) and non-metric multi-dimensional scaling (NMDS); (3) the analysis of similarity (ANOSIM); (4) the unweighted-pair group method with arithmetic mean (UPGMA) clustering; (5) the linear discriminant analysis (LDA) Effect Size (LEfSe) algorithm version 1.1.0 (https://huttenhower.sph.harvard.edu/lefse/) (Segata et al., 2011); and (6) the canonical correlation analysis (CCA) in the PAST software version 4.03 (https://palaeo-electronica.org/2001_1/past/issue1_01.htm) (Hammer, Harper & Ryan, 2001). The details of these analyses were described in the previous article (Pongsilp & Nimnoi, 2024).

Metagenome data analyses

The PICRUSt2 software version 2.5.0 (https://github.com/picrust/picrust2) (Douglas et al., 2020) was employed to analyze the metagenome data for annotation of genes, enzymes, and metabolic pathways by using the information in the Kyoto Encyclopedia of Genes and Genomes (KEGG) (https://www.genome.jp/kegg/) and MetaCyC (https://metacyc.org/) databases. Heat map charts were created to illustrate the abundance and distribution. The PCA in the R software version 4.3.2 (https://www.R-project.org/) (R Core Team, 2023) was performed to display the clusterings of the annotated genes, enzymes, and metabolic pathways among different samples. The associations of food nutrition facts with bacterial communities were analyzed based on Spearman’s correlation coefficients. Statistical significance was indicated by p-value (very significant if p-value was < 0.01 and significant if p-value was < 0.05). The food nutrition facts, the alpha and beta diversity indices as well as the top ten most abundant bacterial classes were included in the Analysis of Variance (ANOVA) using the Tukey’s test. Between-group analysis, ANOVA, and Spearman’s correlation coefficients were computed by the SPSS software version 19.0 (IBM Corp., Chicago, IL, USA). All data analyses were performed with three replicate samples.

Results

Food nutrition facts

All except one of the food nutrition parameters, including calories, carbohydrate, fat, moisture, protein, and sodium contents, were significantly different (p < 0.05) among five Thai traditional fermented foods, while total sugar was only detected in sample A (sweet fermented sticky rice) (Table 1). Glucose was almost the sole (>98.3%) sugar in sample A, while maltose presented at a very low concentration (<1.7%). Sample A contained the significantly highest calories, carbohydrate, and total sugar contents and the significantly lowest moisture and sodium contents. Sample B (sour salt-fermented mustard greens) exhibited the significantly highest moisture content and the significantly lowest calories, fat, and protein contents. Sample C (sour salt-fermented bamboo sprouts) had moderate contents of all food nutrition parameters. Sample D (sour salt-fermented pork) exhibited the significantly highest fat content and the significantly lowest carbohydrate content. Both protein and sodium contents were significantly highest in sample E (sour salt-fermented fish).

Table 1 Food nutrition parameters in five Thai traditional fermented foods.

Food nutrition
parameter	Fermented food sample*	
A	B	C	D	E	
Calories (Kcal/100 g)	181.05 ± 0.02e**	21.62 ± 0.05a	26.32 ± 0.01b	102.61 ± 0.33c	113.47 ± 0.02d	
Carbohydrate (g/100 g)	40.21 ± 0.02e	3.68 ± 0.01b	4.45 ± 0.02c	3.56 ± 0.06a	7.33 ± 0.05d	
Fat (g/100 g)	0.35 ± 0.00c	0.14 ± 0.00a	0.25 ± 0.00b	2.92 ± 0.02e	2.03 ± 0.03d	
Moisture (g/100 g)	55.08 ± 0.09a	92.06 ± 0.03e	81.41 ± 0.02d	73.46 ± 0.03c	68.98 ± 0.03b	
Protein (g/100 g)	4.26 ± 0.01c	1.33 ± 0.01a	1.67 ± 0.01b	15.61 ± 0.15d	16.51 ± 0.12e	
Total sugar (g/100 g)	29.08 ± 0.25b	0.00 ± 0.00a	0.00 ± 0.00a	0.00 ± 0.00a	0.00 ± 0.00a	
Sodium (mg/kg)	69.36 ± 0.69a	7,817.96 ± 1.31c	1,828.56 ± 1.71b	14,054.48 ± 1.50d	15,738.20 ± 2.23e	
Note:

* All values are the means from three replicates ± SD.

** Values with the same letter within a row are not significantly different according to the Tukey’s test.

Sample codes: A, sweet fermented sticky rice (Khao-Mak); B, sour salt-fermented mustard greens (Pak-Kard-Dong); C, sour salt-fermented bamboo sprouts (Nor-Mai-Dong); D, sour salt-fermented pork (Moo-Som); and E, sour salt-fermented fish (Pla-Som).

Sequence analysis and indices of bacterial diversity and richness

Totals of 1,516,672 qualified tags and 4,324 ASVs were obtained from all samples, with a mean Good’s coverage of 99.90 ± 0.17. The Venn diagram (Fig. 1) illustrates the numbers of common, overlapping, and unique ASVs among all foods. Five Thai traditional fermented foods shared ten common ASVs. Samples C (sour salt-fermented bamboo sprouts) showed the highest unique ASVs, followed by samples B, D, E, and A, respectively. Sour salt-fermented vegetables (samples B and C) shared the highest common ASVs. Sample A (sweet fermented sticky rice) was obviously distinct as it shared either none or few common ASVs with the other samples.

Figure 1 Venn diagram presenting the numbers of common, overlapping, and unique ASVs among five Thai traditional fermented foods.

Sample codes (A–E) are as those in Table 1 footnote.

The parameters relative to alpha diversity were evaluated (Table 2). Higher Shannon-Weaver and Simpson’s indices represent greater bacterial diversity. Shannon-Weaver and Simpson’s indices were significantly ranked (p < 0.05), from highest to lowest, as follows: (1) sample C; (2) samples B and E; (3) sample D; and (4) sample A. Bacterial richness indices (Chao1 index and number of observed species) were significantly ranked, (p < 0.05), from highest to lowest, as follows: (1) sample C; (2) sample B; (3) samples D and E; and (4) sample A. These results imply that the bacterial communities of sour salt-fermented vegetables had higher richness than sour salt-fermented meats. Sweet fermented sticky rice possessed the bacterial community with the lowest diversity and richness. On the contrary, the significantly highest dominance was obtained from sample A, followed by samples D, B, E, and C, respectively.

Table 2 Bacterial diversity and richness indices of five Thai traditional fermented foods.

Index	Fermented food sample*	
A	B	C	D	E	
Shannon-Weaver	1.88 ± 0.19a**	4.12 ± 0.06c	7.57 ± 0.30d	3.51 ± 0.02b	4.15 ± 0.06c	
Simpson’s	0.65 ± 0.05a	0.89 ± 0.01c	0.96 ± 0.00d	0.83 ± 0.00b	0.89 ± 0.00c	
Chao1	32.08 ± 5.34a	285.50 ± 54.62c	1879.46 ± 166.01d	129.71 ± 22.66b	150.59 ± 17.85b	
Number of observed species	31.00 ± 4.35a	261.33 ± 43.87c	1769.33 ± 150.44d	128.00 ± 20.88b	143.66 ± 5.71b	
Dominance	0.34 ± 0.05e	0.13 ± 0.01c	0.03 ± 0.00a	0.16 ± 0.00d	0.10 ± 0.00b	
Note:

* All values are the means from three replicates ± SD.

** Values with the same letter within a row are not significantly different according to the Tukey’s test.

Sample codes (A–E) are as those in Table 1 footnote.

Illumina NGS output and bacterial community structures

Among the top ten most abundant phyla present in five Thai traditional fermented foods, the Firmicutes was mostly abundant, ranging between 8.79% and 89.81%, followed by Proteobacteria (7.97–60.75%), Cyanobacteria (0.21–28.88%), Halobacterota (0.00–24.07%), Bacteroidota (0.00–17.00%), Actinobacteriota (0.08–3.59%), Fusobacteriota (0.00–2.89%), Deinococcota (0.00–1.39%), Nanohaloarchaeota (0.00–0.91%), and Campylobacterota (0.00–0.75%). Their relative abundance (Fig. 2) obviously shows that the Firmicutes accounted for the largest proportions in samples D, E, and A, in descending order. The Proteobacteria occupied the largest proportions in samples B and C, with their percentage being much higher in sample B than in sample C. The proportions of Cyanobacteria in sample B and Halobacterota in sample C were remarkably larger than those in the other samples.

Figure 2 Relative abundance of the top ten most abundant phyla present in five Thai traditional fermented foods.

Sample codes (A–E) are as those in Table 1 footnote.

The ASV percentages of the top ten most abundant bacterial classes present in five Thai traditional fermented foods were computed (Table 3). The four classes present in all samples were Bacilli, alpha-Proteobacteria, gamma-Proteobacteria, and Cyanobacteriia. The Bacilli constituted the largest proportions in samples A, D, and E. The alpha-Proteobacteria and gamma-Proteobacteria occupied the largest proportions in samples B and C, respectively. Sample C was the only one whose bacterial community structure consisted of all of the top ten most abundant classes. The Halobacteria, Deinococci, and Nanosalinia were present only in sample C and could be considered as biomarkers of sample C. The bacterial community structures of samples D and E were similar as they consisted of the same seven classes and the Bacilli were the sole major occupant.

Table 3 Percentages of the top ten most abundant bacterial classes present in five Thai traditional fermented foods.

Bacterial class	Fermented food sample*	
A	B	C	D	E	
Bacilli	45.50 ± 5.79b**	8.77 ± 0.81a	8.74 ± 0.43a	89.79 ± 0.97d	73.56 ± 1.66c	
alpha-Proteobacteria	7.82 ± 3.98d	36.73 ± 3.09e	5.17 ± 0.10c	0.35 ± 0.20b	0.21 ± 0.02a	
gamma-Proteobacteria	0.15 ± 0.02a	24.02 ± 0.60d	31.99 ± 2.12e	7.98 ± 0.70b	21.03 ± 1.19c	
Cyanobacteriia	7.53 ± 3.61d	28.88 ± 3.94e	3.23 ± 0.15c	1.47 ± 0.60b	0.21 ± 0.01a	
Halobacteria	0.00 ± 0.00a	0.00 ± 0.00a	24.07 ± 5.08b	0.00 ± 0.00a	0.00 ± 0.00a	
Bacteroidia	0.00 ± 0.00a	0.20 ± 0.07c	15.79 ± 2.31e	0.53 ± 0.01d	0.09 ± 0.01b	
Fusobacteriia	0.00 ± 0.00a	0.00 ± 0.00a	0.03 ± 0.00c	0.01 ± 0.00b	2.89 ± 0.56d	
Actinobacteria	0.00 ± 0.00a	1.29 ± 0.3d	3.41 ± 0.1e	0.12 ± 0.0c	0.07 ± 0.01b	
Deinococci	0.00 ± 0.00a	0.00 ± 0.00a	1.39 ± 0.07b	0.00 ± 0.00a	0.00 ± 0.00a	
Nanosalinia	0.00 ± 0.00a	0.00 ± 0.00a	0.91 ± 0.26b	0.00 ± 0.00a	0.00 ± 0.00a	
Note:

* All values are the means from three replicates ± SD.

** Values with the same letter within a row are not significantly different according to the Tukey’s test.

Sample codes (A–E) are as those in Table 1 footnote.

The finer identification of the unique bacterial community characteristics in five Thai traditional fermented foods was elucidated by a biomarker analysis. The LDA scores from the output of a biomarker analysis, which indicated significant differences in intra-group variation, are shown in Fig. S1. The biomarker genera were totally different among all foods. The Limosilactobacillus was significantly more abundant in sample A. The abundance of the Sphingomonas, Methylobacterium-Methylorubrum clade, and Pediococcus was significantly greater in sample B. The Flectobacillus, Lactobacillus, Janthinobacterium, Halorubrum, Haloplanus, and Halobellus exhibited significantly higher abundance in sample C. The significantly higher abundance of Lactococcus was observed in sample D. The more abundant community in sample E consisted of Companilactobacillus, Proteus, Lactiplantibacillus, Vibrio, Leuconostoc, and Levilactobacillus.

The bacterial genera distribution and abundance in five Thai traditional fermented foods were derived from a heat map analysis. The colors in a heat map chart signify the abundance levels (Fig. 3). A color range from deep blue to dark red denotes the ascending levels of relative abundance. The compositions of dominant genera in each food were remarkably unique. The Limosilactobacillus was the only dominant genus in sample A. The dominant genera in sample B included Bacillus, Pediococcus, Enterococcus, Methylobacterium-Methylorubrum clade, Sphingomonas, and Allorhizobium-Neorhizobium-Pararhizobium-Rhizobium clade. The dominant genera in sample C were Lactobacillus, Deinococcus, Halapricum, Halomicroarcula, Natronomonas, Haloplanus, Janthinobacterium, Halobellus, Flectobacillus, and Halorubrum. Sample D was dominated by Lactococcus and Macrococcus. Sample E was dominantly occupied by Levilactobacillus, Salinivibrio, Latilactobacillus, Shewanella, Leuconostoc, Fusobacterium, Proteus, and Psychrilyobacter.

Figure 3 Heat map chart displaying the bacterial genera distribution and abundance in five Thai traditional fermented foods.

Sample codes (A–E) are as those in Table 1 footnote.

The ANOSIM method indicated that the variations of inter-group bacterial community structure were larger than those of inner-group (R = 1). The pairwise dissimilarity coefficients between sample pairs were quantified and are illustrated in a dissimilarity heat map (Fig. 4A). The colors and dissimilarity coefficient values in a dissimilarity heat map signify the dissimilarity levels of bacterial community compositions between sample pairs. A color range from red to yellow and low-to-high dissimilarity coefficient values denote the ascending levels of dissimilarity. Among sample pairs, the most similar bacterial community compositions were obtained from a sample pair D and E (a dissimilarity coefficient value of 0.124) whereas the most dissimilarity was noticed between samples A and C (a dissimilarity coefficient value of 0.686). These results were additionally confirmed by the clustering analysis for determining the similarity among all samples, which was calculated by the UPGMA method (Fig. 4B). Samples D and E harbored the most similar bacterial communities. Both PCoA and NMDS analyses also provided the consistent results with those of the pairwise dissimilarity and clustering analyses. The ordinations of samples by PCoA and NMDS are illustrated in Fig. 5. Even though the bacterial communities were different among all samples, those of samples D and E were plotted adjacent to each other and alienated from the remaining samples, implying the most similar bacterial communities. On the contrary, the farthest distance was between samples A and C, representing the most dissimilarity in bacterial community.

Figure 4 Dissimilarity heat map and UPGMA dendrogram of the bacterial communities in five Thai traditional fermented foods.

(A) Dissimilarity heat map presenting the dissimilarity coefficient values of sample pairs. (B) UPGMA dendrogram presenting the phyla relative abundance. Sample codes (A–E) are as those in Table 1 footnote.

Figure 5 PCoA and NMDS ordinations of the bacterial composition similarity among five Thai traditional fermented foods.

Sample codes (A–E) are as those in Table 1 footnote. Numbers 1–3 represent sample replicates.

Effects of food nutrition parameters on bacterial community structures

The effects of food nutrition parameters on bacterial community structures in five Thai traditional fermented foods were evaluated (Table S1). The results indicate that the bacterial diversity and richness as well as the classes gamma-Proteobacteria, Bacteroidia, and Actinobacteria were very significantly (p < 0.01) positively associated with moisture content while very significantly negatively associated with calories and total sugar contents. The bacterial diversity was significantly (p < 0.05) negatively associated with fat and protein contents. The bacterial richness was significantly (p < 0.05) negatively associated with fat content. The Bacilli and alpha-Proteobacteria had very significantly (p < 0.01) positive and negative associations, respectively, with fat and protein contents. The Cyanobacteriia were very significantly negatively associated with fat, protein, and sodium contents. The Deinococci were very significantly positively associated with moisture content while very significantly negatively associated with calories, fat, and protein contents.

To definitely reveal the relationships between food nutrition parameters and bacterial communities, the CCA diagram was constructed (Fig. S2). The results show that moisture content influenced the bacterial diversity and richness as well as the classes Halobacteria, Bacteroidia, Deinococci, and Nanosalinia. Calories, carbohydrate, and total sugar contents were directly related to the alpha-Proteobacteria, gamma-Proteobacteria, and Cyanobacteriia. Protein, fat, and sodium contents were directly related to the Bacilli and Fusobacteriia. In order to comprehend the effects of food types on dominant bacterial taxa, ternary plots were depicted to astutely distinguish the relative abundance of the top ten most abundant classes (Fig. 6). As depicted in Fig. 6A, the Bacilli were most closely associated with sample A. The alpha-Proteobacteria and Cyanobacteriia were most closely associated with sample B whereas the Halobacteria and Bacteroidia were most closely associated with sample C. The Actinobacteria were also closely associated with sample C, although their relative abundance was lowest. The gamma-Proteobacteria were dominant across samples B and C. As depicted in Fig. 6B, the Bacilli were dominant across samples A, D, and E whereas the gamma-Proteobacteria were most closely associated with sample E, although their relative abundance was very low.

Figure 6 Ternary plots comparing the bacterial classes and their relative abundance in three out of five Thai traditional fermented foods.

(A) Comparison among plant-derived foods (samples A, B, and C). (B) Comparison among sweet fermented sticky rice and sour salt-fermented meats (samples A, D, and E) Color and size of each circle symbolize the bacterial class and its relative abundance, respectively. Sample codes (A–E) are as those in Table 1 footnote.

Gene functions, enzymes, and metabolic pathways annotated from bacterial communities

Gene functions, enzymes, and metabolic pathways were annotated from the metagenome data of the bacterial communities in five Thai traditional fermented foods against the KEGG and MetaCyC databases using the PICRUSt2 software. The Venn diagrams (Fig. S3) illustrate the numbers of common, overlapping, and unique annotated genes, enzymes, and metabolic pathways among all foods. As depicted in Fig. S3A, a total of 4,448 genes were annotated across all foods, of which 2,636 were common. Sample C had the highest number of unique annotated genes (403) whereas sample A had no any unique annotated gene. For enzyme annotation (Fig. S3B), there were a total of 1,288 annotated enzymes across all foods, of which 889 were common. Sample C also had the highest number of unique annotated enzymes (61) whereas samples A and D had no any unique annotated enzyme. For metabolic pathway annotation (Fig. S3C), there were a total of 63 annotated metabolic pathways across all foods, of which 28 were common. Sample C also possessed the highest number of unique annotated metabolic pathways (20) whereas samples A, D, and E had no any unique annotated metabolic pathway. Seven annotated metabolic pathways were overlapping between both sour salt-fermented vegetables (samples B and C), while in contrast, none were overlapping between both sour salt-fermented meats (samples D and E). There was no overlapping annotated enzyme and metabolic pathway between sample A and either of the remaining samples.

The data of the annotated genes and their relative abundance are displayed in a heat map chart (Fig. 7), in which the colors signify the abundance levels. The color shades ranging from deep blue into yellow, brown, and red symbolize the lowest to highest abundance levels. Gene functions were annotated using the KEGG database (https://www.genome.jp/kegg/). The more abundant genes in sample A were a ligase gene (gshA) (K01919), genes involved in amino acid transport systems (K02029, K02030, and K03293), and transposase genes (K07491 and K07496). The more abundant genes in sample B were a glutathione S-transferase gene (gst) (K00799) and a serine/threonine kinase gene (K08884). The more abundant genes in sample C included genes coding for a reductase (K00059), an outer membrane receptor protein (K02014), and a chemotaxis protein (K03406). The more abundant genes in sample D included genes involved in ATP-binding cassette (ABC) transport systems (K01990, K01992, and K02003), a protease gene (K07052), and a regulatory gene (spxA) (K16509). The sole most abundant gene in sample E was a sugar permease gene (bglF) (K02757).

Figure 7 Heat map chart displaying the genes (KEGG Orthology no.) annotated from the bacterial communities in five Thai traditional fermented foods and their relative abundance.

Sample codes (A–E) are as those in Table 1 footnote.

Enzyme annotation is illustrated in a heat map chart (Fig. 8). The color symbolization is same as described above in Fig. 7. Enzymes were annotated using the KEGG Enzyme database (https://www.genome.jp/kegg/annotation/enzyme.html). The more abundant enzymes in sample A included a ligase (EC:6.3.2.2), a carbamoyl-phosphate synthase (EC:6.3.5.5), and a carboxylase (EC:6.4.1.2). The more abundant enzymes in sample B were a glutathione S-transferase (EC:2.5.1.18) and a histidine kinase (EC:2.7.13.3). The more abundant enzymes in sample C were a cytochrome c oxidase (EC:1.9.3.1) and a peptidylprolyl isomerase (EC:5.2.1.8). The more abundant enzymes in sample D included a DNA methyltransferase (EC:2.1.1.72) and a beta-glucosidase (EC:3.2.1.86). The sole most abundant enzyme in sample E was an RNA helicase (EC:3.6.4.13).

Figure 8 Heat map chart displaying the enzymes (KEGG Enzyme commission no.) annotated from the bacterial communities in five Thai traditional fermented foods and their relative abundance.

Sample codes (A–E) are as those in Table 1 footnote.

Metabolic pathway analysis was elucidated and is displayed in a heat map chart (Fig. 9). The color symbolization is same as described above in Fig. 7. Metabolic pathways were annotated against the MetaCyC database (https://metacyc.org/). Sample A exhibited high abundance of the pentose phosphate (PP) pathways (NONOXIPENT-PWY and PENTOSE-P-PWY) and nucleotide biosynthesis (PWY-7228). Samples B and C were similar regarding that the sole most abundant pathway was aerobic respiration (PWY-3781), though its abundance was strikingly higher in sample C. The more abundant pathways in sample D could be categorized into three groups including (1) glycolysis and organic acid fermentation (ANAEROFRUCAT-PWY, ANAGLYCOLYSIS-PWY, GLYCOLYSIS, and PWY-5484); (2) sucrose degradation (PWY-621); and (3) nucleotide biosynthesis (PWY-7220 and PWY-7222). The more abundant pathways in sample E could be categorized into two groups including (1) glycolysis and organic acid fermentation (ANAEROFRUCAT-PWY, GLYCOLYSIS, PWY-5100, and PWY-5484); and (2) sucrose degradation (PWY-621). Although the pathways in glycolysis, organic acid fermentation, and sucrose degradation were more abundant in samples D and E, their abundance was strikingly lower in sample E than in sample D.

Figure 9 Heat map chart displaying the metabolic pathways (BioCyc ID) annotated from the bacterial communities in five Thai traditional fermented foods and their relative abundance.

Sample codes (A–E) are as those in Table 1 footnote.

Furthermore, ordinations of samples by PCA based on the genes, enzymes, and metabolic pathways annotated from the bacterial community compositions of five Thai traditional fermented foods are depicted in Fig. 10 (A, B, and C, respectively). All three clustering plots consistently attest that the bacterial community compositions of all foods were completely alienated from each other, implying their unique patterns. Sour salt-fermented meats (samples D and E) were the closest pair whereas sour salt-fermented vegetables (samples B and C) and sweet fermented sticky rice (sample A) possessed their own distinct patterns.

Figure 10 Principal component analysis (PCA) plots based on the bacterial community compositions of five Thai traditional fermented foods.

(A) Clustering of the annotated genes. (B) Clustering of the annotated enzymes. (C) Clustering of the annotated metabolic pathways. Sample codes (A–E) are as those in Table 1 footnote. Numbers 1–3 represent sample replicates.

Discussion

This study illustrates the variations in food nutrition as well as bacterial diversity and community among five Thai traditional fermented foods. The results exhibit that sample A (sweet fermented sticky rice) had the significant highest calories and carbohydrate contents and it was the only fermented food in which total sugar was detected. This was possibly related to high starch contents, which typically account for 80–90% of rice grains (Alhambra et al., 2019). Sweet fermented sticky rice also had mildly alcoholic and sour flavors. In general, bacterial and fungal enzymes, such as alpha-amylase and glucoamylase, hydrolyze starch in sticky rice into sugar which is a main product and the fermentation process also yields other organic compounds such as alcohol and lactic acid (Mongkontanawat & Lertnimitmongkol, 2015; Rittisorn et al., 2024). Sample D (sour salt-fermented pork) had the significantly highest fat content. This might due to the original fat content which accounts for 6.3% of pork muscle tissues (Yi et al., 2023). Sample E (sour salt-fermented fish) had the significantly highest protein and sodium contents. Protein detected in sample E might be the remainder of the original protein content as fishes contain protein contents varying from below 10% to over 20% (Alp-Erbay & Yesilsu, 2021).

The results exhibit that samples C (sour salt-fermented bamboo sprouts) and A (sweet fermented sticky rice) had the significantly highest and significantly lowest bacterial diversity and richness, respectively. The dynamic succession of the bacterial community and richness of these kinds of fermented foods has been previously studied using high-throughput sequencing analysis. The bacterial community in sour salt-fermented bamboo shoots was significantly altered during the fermentation time and also correlated with the production of off-odor compounds. The environmental factors affecting the bacterial distribution included salt concentration as well as fermentation time and temperature (Hu et al., 2023). The fermentation of bamboo shoots caused changes in several aspects including food nutrition facts, acidity, toxicity, and organoleptic properties (Hu et al., 2023; Singhal, Satya & Naik, 2021). Rice varieties affected the bacterial diversity in a Chinese traditional rice-based fermented food (Cai et al., 2021). Ethnic tribes of food producers affected the bacterial diversity in Indian traditional rice-based fermented beverages (Yumnam, Hazarika & Sharma, 2024).

This study reports the dominance of bacterial phyla and classes in five Thai traditional fermented foods. The phylum Firmicutes constituted the largest proportions in samples D, E, and A, in descending order. The Proteobacteria were mostly dominant in samples B and C in which their percentage in sample B was much higher than in sample C. The Cyanobacteria and Halobacterota were the second dominant phyla in samples B and C, respectively. These results are similar to those of previous reports. The Firmicutes (Firmicuteota) were mostly dominant in all five samples (>50%) collected from different points along the production line of a Chinese sticky rice fermented sweet dumplings (Suo et al., 2023) and Indian traditional fermented pork fat (De Mandal et al., 2018). The Proteobacteria and Firmicutes established the core microbiotas of Chinese salt fermented mustard greens (Sarengaowa et al., 2024; Wang et al., 2022a), a Taiwanese sour salt-fermented mustard pickle (Chien et al., 2023), and Chinese sour salt-fermented bamboo shoots (Hu et al., 2023).

A heat map analysis was performed to identify the bacterial genera and their relative abundance in five Thai traditional fermented foods. The Limosilactobacillus (formerly Lactobacillus) was the sole dominant genus with extreme relative abundance in sample A (sweet fermented sticky rice). The bacterial community in sweet fermented sticky rice has not been reported elsewhere, though Lactobacillus was one of the dominant members in Chinese sticky rice fermented wines and sweet dumplings (Liang et al., 2020; Suo et al., 2023; Zhao et al., 2022; Zou et al., 2023). The dominant genera in sample B (sour salt-fermented mustard greens) included Bacillus, Pediococcus, Enterococcus, Methylobacterium-Methylorubrum clade, Sphingomonas, and Allorhizobium-Neorhizobium-Pararhizobium-Rhizobium clade. The other studies have found that the LAB (Lactobacillus, Pediococcus, and Weissella), Cobetia, and Halomonas were dominant in Chinese and Thai fermented mustard greens and their bacterial communities were region-dependent (Sarengaowa et al., 2024; Yongsawas et al., 2022). The dominant genera in sample C (sour salt-fermented bamboo sprouts) were Lactobacillus, Deinococcus, Halapricum, Halomicroarcula, Natronomonas, Haloplanus, Janthinobacterium, Halobellus, Flectobacillus, and Halorubrum, while the LAB (Lactobacillus, Lactococcus, and Weissella) were dominant in Chinese sour salt-fermented bamboo shoots (Hu et al., 2023). The Lactobacillus was the sole dominant genus with extreme relative abundance (91.64%) at the day 3 of the fermentation of Indian fermented bamboo shoots (Deka et al., 2021). Sample D (sour salt-fermented pork) contained only two dominant genera including Lactococcus and Macrococcus. Four LAB genera (Lactobacillus, Lactococcus, Pediococcus, and Weissella) established the core bacteriota (>90%) during the fermentation of the other Thai traditional fermented pork (Nham) (Santiyanont et al., 2019). Chinese sour salt-fermented pork contained two dominant LAB (Lactobacillus and Weissella) (Lv et al., 2021). The Lactobacillus constituted almost all of the bacterial communities (96.4–99.9%) in three kinds of Korean salt fermented pork sausages (Kim et al., 2022). The Clostridium accounted for the vast majority (72.48%) at the day 3 of the fermentation of Indian pork fat (Deka et al., 2021). Sample E (sour salt-fermented fish) had the dominant community comprising of Levilactobacillus, Salinivibrio, Latilactobacillus, Shewanella, Leuconostoc, Fusobacterium, Proteus, and Psychrilyobacter. The Turicibacter, Pseudonocardia, Ancylobacter, Gallicola, and Leucobacter exhibited the most abundance (in descending order) in the other kind of Thai traditional salt fermented fish (Pla-Ra) (Phuwapraisirisan et al., 2024). A difference in bacterial community was found between different stages in the salt fermentation of hilsa fish (Tenualosa ilisha). The Enterobacter was mostly abundant in an initial stage (37%), then its abundance was dramatically decreased in a ripe stage (4%). The Cohnella and Bacillus became the two most abundant genera in a ripe stage, accounting for 11% and 10%, respectively (Sarkar et al., 2024). Overall, our current study shows that the LAB were the major bacteria in five Thai traditional fermented foods. The dominant genera in five Thai traditional fermented foods, including Bacillus, Enterococcus, Lactobacillus, Lactococcus, Latilactobacillus, Leuconostoc, Levilactobacillus, Limosilactobacillus, and Pediococcus, have been identified as the LAB (Abedi & Hashemi, 2020; Wang et al., 2021a). The LAB have regularly been recognized as the critical fermentative bacteria in all kinds of fermented foods. Their metabolisms cause the conversion of carbohydrate into lactic acid and impart particular organoleptic properties (Sionek et al., 2023). The LAB ensure the safety of fermented foods by producing organic acids, mainly lactic acid, and a variety of antimicrobial agents which exclude pathogenic microorganisms, and also promote the nutritional value by producing a variety of health-beneficial compounds (Ayed et al., 2024). Differences in bacterial composition among fermented foods might be due to various factors including fermentation process, temperature and time, equipment, geographical origin and type of raw material, water activity, pH value, nutrient availability, humidity, and environmental contamination during storage and sale (Chien et al., 2023; Kim et al., 2022; Zhu et al., 2018).

In this study, the correlations between food nutrition parameters and bacterial communities were evaluated. Our results attest that moisture content was the factor determining the bacterial diversity and richness as well as the abundance of gamma-Proteobacteria, Bacteroidia, Actinobacteria, Deinococci, and Nanosalinia. Fat content was directly related to the abundance of Bacilli. Both protein and sodium contents directly promoted the abundance of Bacilli and Fusobacteriia. The correlations between food nutrition parameters and specific bacterial taxa in the other fermented foods have been reported. The Bacillus, Enterococcus, Lactobacillus, Nocardiopsis, Pediococcus, and Weissella were significantly correlated with various volatile compounds (VCs) in Chinese sweet fermented sticky rice wines (Zou et al., 2023). The LAB (Lactobacillus, Lactococcus, Leuconostoc, and Pediococcus) were correlated with ethanol in a Chinese fermented sticky rice wine (Jiang et al., 2020). The bacterial community structures had positive correlations with color and texture of Chinese traditional salt fermented mustard greens (Sarengaowa et al., 2024). The phylum Firmicutes as well as the genera Enterobacter and Lactococcus were conducive to the production of flavor compounds, while the phyla Cyanobacteria and Proteobacteria affected the texture formation of Chinese traditional sour salt-fermented bamboo shoots (Long et al., 2023). The LAB (Lactobacillus and Weissella) were positively correlated with most of VCs and free amino acids (FAAs) in Chinese sour salt-fermented pork (Lv et al., 2021). The microbial community structure of Chinese traditional salt fermented pork was affected by pH value, water activity, NaCl, and total volatile basic nitrogen (TVB-N) (Wang et al., 2021b). The Bacillus, Gallicola, Proteiniclasticum, and Pseudonocardia were positively correlated with sweet, cheesy, soy sauce-like, and fish sauce-like aromas of the other kind of Thai traditional salt fermented fish (pla-ra) (Phuwapraisirisan et al., 2024). The LAB (Lactobacillus, Lactococcus, and Leuconostoc), Brochothrix, and Providencia played a key role in the production of esters which were major flavor compounds, while the Providencia, Vagococcus, and Weissella were alcohol producers in Chinese traditional salt and chili fermented fish (Yin et al., 2024). The LAB (Lactococcus and Latilactobacillus) exerted the highest influences on the production of most volatile flavor compounds in Chinese salt and sugar fermented tilapia fish surimi (Li et al., 2024).

The comprehensive and quantitative analyses were performed to compare the abundance of genes, enzymes, and metabolic pathways annotated from the bacterial communities in five Thai traditional fermented foods. Genes, enzymes, and metabolic pathways were indirectly annotated from the obtained bacterial 16S rRNA gene sequence data using the PICRUSt2 software. Even though the annotation can be achieved by utilizing the genome data available on the KEGG and MetaCyC databases, the confinement was earlier stated. The existing reference genome data on the databases has a tendency to influence the annotation, although this constraint has been diminished as more highly-qualified genome data is accessible (Douglas et al., 2020; Wright & Langille, 2025). The annotation is also insufficient to distinguish gene functions which are specific to strains (Douglas et al., 2020). Our results unveil that the abundant genes, enzymes, and metabolic pathways of all foods were those involved in growth metabolisms, genetic information processing, environmental information processing, and cellular signaling. The PP pathways were mostly abundant in sweet fermented sticky rice (sample A) and their abundance was strikingly higher than in the other foods. The PP pathways are crucial to carbon-balancing processes, adaptation to oxidative stress as well as biosyntheses of amino acids and nucleotides (Stincone et al., 2015). During the fermentation process, the PP pathways are vital to provide nicotinamide adenine dinucleotide phosphate (NADPH) for various metabolisms including the production of alcohol and organic acids (Laëtitia, Pascal & Yann, 2014; Masi, Mach & Mach-Aigner, 2021), as evidenced by the enrichment of the PP pathways in the fermentation of rice bran (Chen & Li, 2023). The other abundant genes, enzymes, and metabolic pathways in sample A included those essential for amino acid metabolism, genetic information processing, cellular signaling as well as biosyntheses of amino acids, fatty acids and nucleotides. Similarly, carbohydrate, sugar, and amino acid metabolisms were the most influential pathways in Chinese fermented rice wines (Jiang et al., 2020; Zhao et al., 2022). The most abundant genes and enzymes in sour salt-fermented mustard greens (sample B) were those of a glutathione S-transferase, which is responsible for cellular detoxification, oxidative stress response, and basal metabolism (Lienkamp et al., 2021), a serine/threonine kinase, which modulates cellular signaling, physiology, proliferation, virulence, and antibiotic persistence (Nagarajan, Lenoir & Grangeasse, 2022), and a histidine kinase, which is connected to cellular signaling, pathogenicity, virulence, persistence, biofilm development, and antibiotic resistance (Ahsan et al., 2024). The sole most abundant pathway in sample B was aerobic respiration for providing energy to cells. As reported in the other study, the pathways in biosyntheses of amino acids and secondary metabolites as well as carbon metabolism, environmental information processing, and cellular signaling were annotated in Chinese salt fermented mustard greens, though their abundance was significantly different among regional samples (Sarengaowa et al., 2024). The most abundant genes in sour salt-fermented bamboo sprouts (sample C) function in amino acid metabolism, fatty acid biosynthesis, genetic information processing, and cellular signaling. The most abundant enzymes in sample C were a cytochrome c oxidase, which is crucial to aerobic respiration (Hederstedt, 2022), and a peptidylprolyl isomerase, which catalyzes protein folding (Anchal, Kaushik & Goel, 2021). As same as sample B, the sole most abundant pathway in sample C was aerobic respiration. The major metabolic pathways in the fermentation of Chinese bamboo shoots were gluconeogenesis, the tricarboxylic acid (TCA) cycle, the PP pathways, carbon fixation as well as metabolisms of vitamin, coenzyme A (CoA), sugar, amino sugar, nucleotide sugar, nucleotide, and methane (Hu et al., 2023). The more abundant genes in sour salt-fermented pork (sample D) were those involved in genetic information processing and cellular signaling as well as a protease gene whose abundance was strikingly higher in sample D than in the other foods. This was possibly related to the high protein content in pork, which has an average of 27.6 g/100 g (Drewnowski, 2024). The more abundant enzymes in sample D included a DNA methyltransferase for regulating a wide range of cellular processes (Gao et al., 2023) and a beta-glucosidase for saccharide hydrolysis. The more abundant pathways in sample D play roles in glycolysis, organic acid fermentation, sucrose degradation, and nucleotide biosynthesis. Energy, carbohydrate, and amino acid metabolisms were functional in Indian traditional fermented pork fat (De Mandal et al., 2018). The sole most abundant gene in sour salt-fermented fish (sample E) was a sugar permease gene (bglF) which plays roles in environmental information processing and cellular signaling. The sole most abundant enzyme in sample E was an RNA helicase for RNA metabolism. The major functional metabolic pathways in sample E were those involved in glycolysis, organic acid fermentation, and sucrose degradation. The metabolic pathways were dynamic over time during the fermentation of a Chinese traditional fish sauce, during which carbohydrate, amino acid, and nucleotide metabolisms gradually dominated whereas energy metabolism gradually abated (Wang et al., 2022b).

The annotated genes, enzymes, and metabolic pathways in fermented foods reflect the influences of food-related factors (e.g., raw materials, food nutrition facts, starter microorganisms, and fermentation environments) on bacterial diversity and richness. Thus, the comprehensive and quantitative study on the bacterial diversity and communities of various kinds of fermented foods will facilitate understanding of the functional roles of microbiomes which contribute to the product quality and successful fermentation.

Conclusion

The present study investigated the bacterial diversity and compositions in five Thai traditional fermented foods. Almost all food nutrition parameters (calories, carbohydrate, fat, moisture, protein, total sugar, and sodium contents) significantly varied across all foods, while total sugar was only detected in sweet fermented sticky rice. Sour salt-fermented bamboo sprouts and sweet fermented sticky rice had the significantly highest and significantly lowest bacterial diversity and richness, respectively. The most abundant phyla present in all foods were Firmicutes, Proteobacteria, Cyanobacteria, and Actinobacteriota, though their proportions were variable in different foods. The bacterial community structures of both sour salt-fermented meats (pork and fish) were proximate to each other while those of sweet fermented sticky rice and sour salt-fermented bamboo sprouts were mostly alienated from each other. The effects of food nutrition parameters on bacterial communities were statistically identified. Moisture and calories contents exerted very significantly positive and negative impacts, respectively, on the bacterial diversity and richness as well as the classes gamma-Proteobacteria, Bacteroidia, Actinobacteria, and Deinococci. Total sugar was very significantly negatively associated with the bacterial diversity and richness as well as the gamma-Proteobacteria, Bacteroidia, and Actinobacteria. Fat and protein contents had very significantly positive associations with the Bacilli and very significantly negative associations with the alpha-Proteobacteria, Cyanobacteriia, and Deinococci. Sodium content very significantly negatively affected the Cyanobacteriia. This study unveils the bacterial microbiomes and keystone taxa in five Thai traditional fermented foods and also profiles the differences in metagenomic and metabolomic compositions for understanding the bacterial succession in the fermentation processes.

Supplemental Information

Supplemental Information 1 LDA scores derived from the output of a biomarker analysis indicating the bacterial taxa whose numbers were significantly different among five Thai traditional foods.

Sample codes (A-E) are as those in Table 1 footnote.

Supplemental Information 2 CCA diagram illustrating the effects of food nutrition parameters on bacterial communities.

Food nutrition parameters influencing bacterial communities are represented by green lines whose lengths represent effect levels.

Supplemental Information 3 Venn diagrams presenting the numbers of common, overlapping, and unique genes, enzymes, and metabolic pathways annotated from the bacterial communities of five Thai traditional fermented foods.

(A) Genes (B) Enzymes (C) Metabolic pathways. Sample codes (A-E) are as those in Table 1 footnote.

Supplemental Information 4 Spearman’s (rs) correlations between nutrition factors and bacterial classes in five Thai traditional fermented foods.

Supplemental Information 5 Raw data.

Supplemental Information 6 Amplicon Sequence Variables (ASVs) raw sequences.

Additional Information and Declarations

Competing Interests

The authors declare that they have no competing interests.

Author Contributions

Pongrawee Nimnoi conceived and designed the experiments, performed the experiments, analyzed the data, prepared figures and/or tables, authored or reviewed drafts of the article, and approved the final draft.

Neelawan Pongsilp conceived and designed the experiments, performed the experiments, analyzed the data, prepared figures and/or tables, authored or reviewed drafts of the article, and approved the final draft.

DNA Deposition

The following information was supplied regarding the deposition of DNA sequences:

The sequences of metagenomics data are available at NCBI: PRJNA1180770.

Data Availability

The following information was supplied regarding data availability:

The raw data of food nutrition parameters, Spearman’s correlations, Bacterial diversity and richness indices, top ten most abundant classes and top ten most abundant phyla are available in the Supplemental File.

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
