# Peer review of "Insights into the metagenomic and metabolomic compositions of the bacterial communities in Thai traditional fermented foods as well as the relationships between food nutrition and food microbiomes"

_PeerJ, doi:10.7717/peerj.19606_

## Round 0.1 · original submission · Major Revisions

Dear Dr. Pongsilp, You see that the reviewers liked your manuscript but they suggested several improvements and revisions which I request you to address carefully.

Best regards,
Elisabeth

Reviewer 1 ·

Basic reporting

The manuscript titled "Insights into the Metagenomic and Metabolomic Compositions of the Bacterial Communities in Thai Traditional Fermented Foods, as well as the Relationships Between Food Nutrition and Food Microbiomes" focuses on the study of five fermented foods, integrating chemical and microbiome data.

Below are my revisions and suggestions:

Line 55: Please provide additional information on lactic acid bacteria (LAB) taxa, supported by relevant references.
Lines 105-114: Did you perform these analyses on three different batches, or were they conducted on a single batch divided into three subsamples? It is preferable to analyze multiple batches to obtain more representative data.
Lines 116-127: Please specify the tools used for analysis, including their versions (e.g., RStudio, PAST3, etc.).
Line 129: Did you apply any additional tools to the FASTQ files before using PICRUSt2?
Line 142: Was ANOVA used for all comparisons? Additionally, how did you assess the parametricity of the data?
Lines 151-154: Significant in comparison to what? Furthermore, how was the statistical analysis conducted if each food type had three technical replicates but no biological replicates?
Line 179: Please simplify this sentence, as the numerical details may not be necessary.

Experimental design

General Comment

Including multiple batches per food type would enhance the robustness of the study.

Validity of the findings

The validity of the finding can be increased including multiple batches per food type would enhance the robustness of the study.

Reviewer 2 ·

Basic reporting

The manuscript was well prepared. Literature and references were enough to describe the study. English was good to explain results.

Experimental design

Methods were well described in materials and methods sections.

Validity of the findings

statistical analysis were performed to obtained data.

Additional comments

This study was well described the microbiota of the five thai fermented foods. The results were well explained and discussed. It contributes the food microbiology research area to describe microbiota of fermented food using next generation sequnecing technique.

·

Basic reporting

pass

Experimental design

pass

Validity of the findings

pass

Additional comments

This study provides valuable insights into Thai fermented foods, but revisions addressing methodological transparency, statistical rigor, and contextual interpretation are needed. With these improvements, the manuscript will significantly contribute to the field of food microbiology.
1. The manuscript is generally well-written, but some sections lack conciseness. For example, the abstract mentions "highest unique ASVs" but does not provide numerical values (e.g., exact ASV counts for sample C).
2. The use of PICRUSt2 for metagenomic inference should acknowledge limitations (e.g., reliance on 16S data vs. shotgun metagenomics). Clarify how these limitations might affect pathway predictions.
3. The ANOVA/Tukey test is appropriate, but p-values alone are insufficient. Include effect sizes to quantify differences in bacterial diversity/nutrition parameters.
4. Figure 1 : Label overlapping ASV numbers more clearly. Consider using UpSet plots for better visualization of complex overlaps. Heatmaps (Figures 3,7–9): Ensure consistent color scales across figures.
6. Compare findings with similar studiesto highlight unique features of Thai fermented foods. Expand on why specific pathways (e.g., PP pathways in “Khao-Mak”) dominate. Do these pathways correlate with flavor/aroma compounds?
7. Table 1: Report units for nutrition parameters and include standard deviations. Define all abbreviations (e.g., LAB, ASV) upon first use.

---

## Round 0.2 · Minor Revisions

Your manuscript has highly improved by the revision. But there are still some issues in the revised manuscript which require your attention. I highlighted some typos and Grammar issues in the attached manuscript. Please carefully revise them and check the entire manuscript for additional ones.

**Language Note:** The Academic Editor has identified that the English language must be improved. PeerJ can provide language editing services - please contact us at [email protected] for pricing (be sure to provide your manuscript number and title). Alternatively, you should make your own arrangements to improve the language quality and provide details in your response letter. – PeerJ Staff

Reviewer 1 ·

Basic reporting

My suggestions have been followed with more details, I have no further revisions.

Experimental design

My suggestions have been followed with more details, I have no further revisions.

Validity of the findings

I have no further revisions, the study design and finding are well reported.

---

## Round 0.3 · Minor Revisions

Dear Dr. Pongsilp,

Thank you for the improvements of your manuscript. However, you corrected only a few of the grammatical and typing errors in your manuscript.

As the academic editor of the manuscript, I request the following:

The manuscript requires substantial improvement in scientific English to meet our publication standards.

Final acceptance is contingent upon meeting these high language requirements.

Thank you for your consideration

Kind regards,
Elisabeth Grohmann

**Language Note:** The Academic Editor has identified that the English language must be improved. PeerJ can provide language editing services - please contact us at [email protected] for pricing (be sure to provide your manuscript number and title). Alternatively, you should make your own arrangements to improve the language quality and provide details in your response letter. – PeerJ Staff

---

## Round 0.4 · accepted · Accept

Thank you for the careful revision of your manuscript. Now, all my comments have been addressed.